# Incidence of Surgical Site Infection Following Cesarean Section and Its Associated Factors in a Hospital of the Eastern Region, Saudi Arabia: A Retrospective Cohort Study

**DOI:** 10.3390/healthcare12151474

**Published:** 2024-07-25

**Authors:** Sukinah F. Albaharnah, Sara A. Rashed, Rahaf S. Almuhaimeed, Salah Abohelaika

**Affiliations:** 1Obstetrics and Gynecology Department, Qatif Central Hospital, Qatif 32654, Saudi Arabia; salbaharnah@moh.gov.sa; 2Faculty of Medicine, Mansoura University, Mansoura 35516, Egypt; sara.alialrashed@gmail.com; 3College of Medicine, Imam Abdulrahman Bin Faisal University, Dammam 34212, Saudi Arabia; dr.rahaf.almuhaimeed@gmail.com; 4Research Department, Qatif Central Hospital, Qatif 32654, Saudi Arabia; 5Pharmacy Department, Qatif Central Hospital, Qatif 32654, Saudi Arabia

**Keywords:** surgical site infections, cesarean section, Saudi Arabia

## Abstract

Background: Surgical site infections (SSIs) following cesarean sections (CS) pose significant challenges in healthcare settings, prompting this five-year investigation in a Qatif Area general hospital. The study aimed to delineate nuances in SSI occurrences, assess yearly variations, and identify associated risk factors impacting SSI rates among CS patients. Methods: A retrospective analysis of 1584 cesarean sections conducted over five years was undertaken, and the reported SSI cases were examined to calculate the overall and yearly SSI rates. The impact of potential risk factors such as diabetes mellitus, hypertension, and postoperative antibiotic courses on SSI occurrence was examined. Results: The study revealed an overall SC rate of 15.4%. The SSI rate following CS was 4.7% (N = 74), with variations across years ranging from 2.2% in 2020 to a peak of 6.7% in 2022. Notably, 2021 and 2022 demonstrated increased SSI rates compared to prior years. Diabetes mellitus and a surgery duration of more than one hour exhibited a substantial association with SSI occurrence, (OR = 10.76, *p* = 0.038) and (OR = 3.54, *p* = 0.002), respectively, signifying independent risk factors. Conclusions: The study underscores the dynamic nature of SSI rates following CS, highlighting an increasing trend in recent years. All cases were managed with the optimal hospital care. Diabetes mellitus and a surgery duration of more than one hour emerged as prominent independent risk factors for SSI, warranting heightened vigilance and tailored preventive measures in this subset of patients.

## 1. Introduction

The global rise in cesarean section (CS) rates over the past thirty years has been a notable trend in obstetric care based on World Health Organization (WHO) reports [1]. Numerous factors contribute to this increase, including changes in medical practices, maternal requests for elective C-sections, advancements in technology, and a higher prevalence of certain risk factors that might necessitate a C-section delivery [2,3,4,5]. This surge in CS rates raises various concerns and prompts extensive research and discussion within the medical community. While C-sections are life-saving procedures when medically indicated, the increase in elective or non-medically necessary surgeries raises questions about potential risks and complications associated with this mode of delivery [6]. Surgical site infections (SSIs) following CS represent a significant concern in the realm of obstetric care and are indeed among the most prevalent hospital-associated infections [7]. Despite the fact that SSIs are the second most common type of healthcare-associated infections (HCAIs) that result in a health financial burden with an annual estimation of USD 3.3 billion in USA, SSIs are the most preventable type of HCAIs [8]. These infections occur at the site of surgery and can lead to complications that impact the health of both the mother and the newborn [9]. The Centers for Disease Control and Prevention (CDC) defines SSI as wound infection within 30 days of operation. SSIs could be further classified into superficial incisional, deep incisional, and organ or space SSIs according to CDC additional severity criteria [1].

SSIs represent a substantial burden on patients and healthcare systems across various types of surgical procedure. The impact of SSIs extends beyond the immediate postoperative period and affects patients’ recovery and overall health outcomes [10,11]. SSIs often lead to extended hospital stays as patients require additional care to manage the infection [12]. This prolonged hospitalization not only affects the patient’s physical recovery but also increases healthcare costs significantly [13]. Treating SSIs requires additional medical resources, including medications, wound care supplies, laboratory tests, and potentially further surgical interventions. The added expenses associated with managing these infections significantly contribute to healthcare costs. In severe cases, SSIs could contribute to patient deterioration, the progression of underlying health conditions, and, in rare instances, mortality [12].

The rates of SSIs following CS vary across diverse global landscapes and the prevalence rates range between 4 and 25% [14,15,16,17]. These variations reflect a multitude of factors, encompassing differences in healthcare infrastructure, infection control practices, patient demographics, antibiotic usage, and reporting methodologies. Addressing this global diversity in SSI rates requires collaborative efforts, standardized protocols, enhanced healthcare resources, and a commitment to sharing best practices internationally. The importance of researching SSIs post C-section lies in the potential risks they pose. Additionally, SSIs can impact the postoperative recovery process, affecting a mother’s ability to care for her newborn and potentially interfering with breastfeeding and bonding [18]. Understanding the incidence and associated risk factors of SSIs after CS play a fundamental role in raising healthcare professionals’ awareness, enabling proactive measures for SSI prevention and subsequently enhancing maternal outcomes. However, within the context of the Eastern Region of Saudi Arabia, a comprehensive investigation into the incidence, risk factors, and management of SSIs among women undergoing CS remains notably absent from existing research. This gap in knowledge presents an opportunity to address a critical aspect of maternal healthcare that has not been thoroughly explored in our region, the Eastern Region of Saudi Arabia. Hence, the primary objective of our study was to meticulously assess the incidence rates, identify prevalent risk factors, and scrutinize the management protocols concerning SSIs following CS procedures. This investigation was conducted over a span of five years (2018–2022) at a secondary hospital situated in Qatif Province, aiming to shed light on the specific nuances of SSI occurrences within this local healthcare setting. By delving into these crucial aspects within the Saudi Arabian context, our study endeavors to contribute essential insights that can potentially inform healthcare practices, refine preventive strategies, and ultimately elevate the standard of maternal care in this region.

## 2. Materials and Methods

This investigation was conducted at Qatif Central Hospital (QCH), a 330-bed hospital in the Eastern Region of Saudi Arabia, using a retrospective dataset on a cohort of women who underwent CS in both emergency and elective settings and encountered SSI within 30 days post CS. The sample size was determined based on the prevalence of SSI among females undergoing CS at QCH from 2018 to 2022 who met specific criteria: SSI occurrence within 30 days post CS in emergency or elective settings, either during the same admission stay or after being readmitted without treatment initiation.

The clinical protocol of our institution stipulates discharge on the third postoperative day following a CS and on the fifth day after gynecological procedures for patients in good postoperative health. A follow-up assessment occurs on the 10th day, while the study’s research protocol extends the follow-up period to 30 days post procedure. Surveillance for nosocomial infections occurs exclusively in the outpatient department within the clinic due to the absence of alternative monitoring programs. Wound cultures were obtained using sterile swabs and submitted to the department of microbiology for culture-based analysis to identify postoperative wound pathogens. All women undergoing operative deliveries were eligible for inclusion in the study, while exclusion criteria encompassed patients initially operated in private hospitals and subsequently hospitalized in the clinic.

### 2.1. SSI Diagnosis Criteria

SSI diagnosis criteria adhere to CDC guidelines [1], requiring the presence of purulent discharge or pus from the incision, accompanied by pain and at least two cardinal symptoms of inflammation.

Data Collection: Data retrieval involved gathering information from patient electronic medical records within the hospital system, as well as documentation from the operation room, labor and delivery, obstetric wards, and intensive care unit from 2018 to 2022. The study did not involve any procedures or interventions.

### 2.2. Data Analysis and Management

The analysis was conducted using SPSS version 26 software. Data management involved the meticulous handling of missing values, outliers, and validation checks to ensure dataset integrity. Descriptive statistics were generated, presenting frequencies and percentages for categorical variables related to CS patients and the occurrences of SSIs. A logistic regression model was conducted to predict the risk factors associated with SSIs among CS patients. Predictor variables were meticulously selected based on a thorough review of the literature and preliminary analyses. Odds ratios (OR), 95% confidence intervals (CI), and corresponding *p*-values were calculated for each predictor variable, enabling the identification of factors significantly associated with an increased or decreased risk of SSIs in CS patients.

## 3. Results

The analysis showed that, over five years, a total of 1584 (15.4%) cesarean sections were conducted at QCH from 10,308 deliveries, resulting in 74 reported cases of SSI, equating to an overall SSI rate of 4.7%. Yearly variations were observed in SSI percentages, ranging from a low of 2.2% in 2020 to a peak of 6.7% in 2022. Specifically, 2021 and 2022 demonstrated an uptick in SSI rates compared to previous years (Table 1).

Among those who had SSI, 60.8% were aged 34 years or below compared to 39.2% of those aged 35 or older. Most patients were of Saudi nationality (87.8%). Body mass index (BMI) indicated that 67.6% had a BMI of 30 or higher. Regarding parity, 83.8% of patients had a parity of 1–4. Gestational age showed that 74.3% of patients had a gestational age of 37 weeks or more. More than half of them (55.4%) experienced labor before the operation, and about 40.5% had a ruptured membrane status. Concerning comorbidities, the most prevalent were diabetes mellitus (25.7%) and hypertension (10.8%), while other conditions like sickle cell disease and cardiovascular disease were less common, at 8.1% and 2.7%, respectively. Approximately half of the patients (48.6%) had no reported comorbidities (Table 2).

In the context of cesarean surgeries, Table 3 outlines various surgical characteristics of the procedures conducted. The majority of operations were classified as emergency cases (71.6%) compared to elective (28.4%). General anesthesia was the commonly employed option (59.5%), followed by spinal anesthesia (39.2%), with regional anesthesia being the least frequent (1.4%). The duration of premature rupture of membranes (PROM) was predominantly not recorded (87.8%), while instances of PROM for less than 6 h accounted for a small percentage (1.4%), and those exceeding 6 h were reported in 10.8% of cases. Antibiotics were primarily administered before the skin incision (97.3%) rather than solely before entering the operating room (2.7%). Blood transfusions were relatively infrequent (12.2%). The postoperative hospital stay was less than 3 days for most patients, while 14.9% had an extended stay of ≥3 days.

The majority of SSIs were superficial (89.2%), while a small percentage were categorized as deep (8.1%) or involving space (2.7%). Regarding treatment, most cases received systemic antibiotics (89.2%), while a few underwent local treatment (6.8%), and only one patient (1.4%) required both systemic antibiotics and surgery (Table 3). Cultures conducted indicated various results: no growth was observed in 29.7% of cases, Gram-positive bacteria were identified in 33.8% of cases, Gram-negative bacteria were found in 25.7% of cases, and mixed growth in 10.8% of cases. Of the total isolates, *Staphylococci* constituted 44.2%, *Streptococci* 23.1%, anaerobes 3.8%, and other Gram-negative species 28.8%.

A logistic regression analysis was employed to investigate risk factors associated with SSIs in cesarean patients (Table 4). The model showed that patients with diabetes and those who had surgery duration ≥ one hour were 10.76 and 3.54 times, respectively, more likely to have SSI [OR = 10.76 (95% CI = 1.14–101.70), *p* = 0.038; OR = 3.54 (95% CI = 1.49–7.17), *p* = 0.002, respectively]. Other factors such as age, nationality, BMI ≥ 30, parity ≥ 4, gestational age, labor before operation, membrane status, sickle cell disease, cardiovascular diseases, no comorbidities, type of operation, anesthesia, duration of PROM, and blood transfusion did not suggest significant influence on SSI risk (*p* > 0.05).

## 4. Discussion

Surgical site infections following CS are a notable concern in obstetric care, and the factors contributing to SSIs in CS patients are multifaceted. They encompass various preoperative, intraoperative, and postoperative elements. Previous reports have shown that many factors play crucial roles in SSI development, which include hypertension, diabetes, the duration of labor, the rupture of membranes, the type of anesthesia, and postoperative care [19,20,21,22,23,24]. Additionally, the emergence of antibiotic-resistant pathogens further complicates infection control and treatment strategies [25]. Our findings showed that diabetes mellitus was an independent risk factor for SSI in our study cohort (OR = 10.76, *p* = 0.038). The findings emphasize the pivotal role of diabetes mellitus as a prominent risk factor for SSIs in CS patients. Diabetes mellitus has been consistently identified as a significant risk factor for SSIs in various surgical procedures other than CS [26]. This association is grounded in the physiological changes and immunological alterations inherent in diabetes that contribute to an increased susceptibility to infections. Elevated blood glucose levels in diabetes impair the function of immune cells, such as neutrophils and macrophages, which is crucial for defense against infections. This impairment compromises the body’s ability to combat pathogens, increasing the risk of SSIs [27]. Diabetes often leads to microvascular complications and impaired wound healing due to reduced angiogenesis and collagen synthesis, fostering an environment conducive to infections [28]. Diabetes can cause changes in the skin’s microbiota, potentially promoting colonization by pathogenic bacteria, predisposing individuals to SSIs [29]. Variations in glycemic control among diabetic patients might affect the degree of susceptibility to SSIs. Those with poorly controlled diabetes might have a higher risk compared to well-managed cases [27]. Variability in diabetes management, duration, severity, and type across studies can lead to disparate results and differing impacts on SSI development [30,31,32]. This underscores the importance of stringent perioperative management and preventive measures in diabetic CS patients to minimize SSI occurrences. Close monitoring, early intervention, and tailored infection control strategies specific to diabetic patients undergoing CS are warranted to mitigate these heightened risks.

CS rates have increased in the past three decades in both developing and developed countries [33]. In Saudi Arabia, the CS rate in 1997 was 10.6%, increased to 19.1% in 2006 [34], and recently reached up to 48.6% in some Saudi health centers [34,35,36,37,38]. In this study, the reported SC prevalence of 15.4% is lower than what has been reported in these studies. On the other hand, the observed SSI rate of 4.7% in this study appears notably low, but there was an increase in SSI rates in 2021 and 2022; however, it aligns with reported global SSI rates following CS. The incidence of SSI in 2020 was quite low compared with the other years. The drop in the rate of SSI in this year probably resulted from the fact that our hospital was a COVID-19 treatment center; some pregnant women infected with COVID-19 were aggressively treated with broad-spectrum antibiotics in the critical care unit, and most of them had their pregnancies terminated by CS to save their lives. For this reason, the rate of CS in 2020 was the highest (20.4%) compared to other years. All of these cases received immediate and extensive hospital care. Across various studies worldwide, SSI rates after CS exhibit substantial variability. For instance, reported rates range from as low as 2.7% in Nova Scotia [39] to 5.5% in the USA [40], contrasting starkly with markedly higher rates of up to 48% in low-resource settings in Tanzanian tertiary hospitals [41], 23.5% in Brazil [42], 18.8% in Malaysia [43], 14.4% in Jordan [44], and 6.2% in an Estonian University Hospital [45]. Another study conducted in the Republic of Kosovo reported a higher prevalence (9.85%) and identified increases in age, previous surgery, preoperative antibiotics, and the duration of surgery as independent risk factors for SSIs [46]. In Saudi Arabia, the reported SSI rates after CS were between 1.37% and 4.5% [4,47,48,49,50]. These studies underscore the wide-ranging variations in SSI rates, influenced by factors such as the study population, underlying health conditions, the administration of antibiotics, and the reliability of methods used for SSI identification and reporting. The observed increase in SSI rates beyond the 1 h duration of surgery suggests a potential relationship between longer procedures and a higher likelihood of surgical site infections. Opøien et al. found that SSI incidence increased by about twofold among CS patients when the operative time was more than 0.6 h, a threshold that is significantly lower than the mean operative time [51]. While the correlation between longer surgeries and higher SSIs is noted, establishing a causal relationship is complex. Other variables, such as patient health status, surgical technique, sterility maintenance, and postoperative care, could also influence SSI rates [52]. Another study also reported that open or laparoscopic surgery increased the incidence of SSIs twofold compared to other surgical techniques [53]. Recognizing the heightened SSI risk in prolonged or high-blood-loss surgeries, our proposed adjustment in antibiotic dosing frequency is tailored to address the unique challenges posed by these procedures. For surgical procedures lasting over 4 h or those associated with blood loss exceeding 1500 mL, we recommend a revised antibiotic prophylaxis regimen. Administering a repeated dose of an antibiotic, for instance, throughout the extended surgical procedure aims to enhance antimicrobial coverage and potentially reduce SSIs [54]. All these findings might prompt healthcare providers to reconsider surgical protocols, operating room practices, or pre-/postoperative measures for cesarean deliveries. Strategies to minimize surgical duration or optimize infection prevention protocols beyond the critical time threshold could be explored to reduce SSI risk. Some quality improvement projects to decrease SSI rates include optimizing pregnant women’s general health from the beginning of pregnancy with early antenatal care, managing chronic diseases that affect pregnancy outcome such as diabetes, and ensuring the approval of any emergency CS by a gynecology consultant whenever possible.

There are some limitations to the current study. Results might not be applicable to a broader population because the current study did not account for other maternal variables such as prenatal care, labor disorders, the rate of gestational diabetes compared to pre-existing diabetes, and the prevalence of urinary tract infections following a CS. Further investigation into these matters could be conducted through a more extensive cohort study.

## 5. Conclusions

The overall surgical site infections in our cohort were found to be 4.7%. Despite variations in the rates over the years, an upward trend in 2021 and 2022 prompts attention to mitigating infection risks in cesarean procedures. Factors such as diabetes mellitus and increased operative duration exhibited a considerable independent association with a heightened SSI risk, emphasizing the need for tailored interventions and vigilant monitoring for this subgroup. Targeted interventions addressing diabetes-related complications, cautious antibiotic management aligned with evidence-based durations, and ongoing surveillance should be implemented to curtail SSIs in cesarean deliveries. Strategies focused on optimizing patient-specific care, especially for those with diabetes, could lead to significant reductions in postoperative complications and enhance overall maternal health outcomes. Continued research and the implementation of tailored interventions guided by these findings are crucial in advancing cesarean care and reducing SSI occurrences in this population.

## Figures and Tables

**Table 1 healthcare-12-01474-t001:** Prevalence of cesarean sections and surgical site infections due to cesarean sections over five years.

Year	Total Deliveries	CS, n (%)	SSI, n (%)
2018	2663	286 (10.7)	14 (4.9)
2019	2133	288 (13.5)	13 (4.5)
2020	1763	359 (20.4)	8 (2.2)
2021	1968	353 (17.9)	19 (5.4)
2022	1781	298 (16.7)	20 (6.7)
**TOTAL**	**10,308**	**1584 (15.4)**	**74 (4.7)**

CS = cesarean section; SSI = surgical site infection.

**Table 2 healthcare-12-01474-t002:** Sociodemographic and pre-existing factors in cesarean patients with SSI (N = 74).

		N	%
**Age**	<34 years	45	60.8
	≥35 years	29	39.2
**Nationality**	Non-Saudi	9	12.2
	Saudi	65	87.8
**Body mass index**	<30	24	32.4
	≥30	50	67.6
**Parity**	1–4	62	83.8
	>4	12	16.2
**Gestational age**	<37 weeks	19	25.7
	≥37 weeks	55	74.3
**Labor before operation**	No	33	44.6
	Yes	41	55.4
**Status membrane**	Intact	44	59.5
	Ruptured	30	40.5
**Comorbidities**	Diabetes mellitus	19	25.7
	Hypertension	8	10.8
	Sickle cell disease	6	8.1
	Cardiovascular disease	2	2.7
	No comorbidities	36	48.6

**Table 3 healthcare-12-01474-t003:** Surgery (cesarean)-related characteristics.

		N	%
Type of operation	Elective	21	28.4
	Emergency	53	71.6
Duration of operation	<1 h	20	27.0
	≥1 h	54	73.0
Type of anesthesia	General	44	59.5
	Spinal	29	39.2
	Regional	1	1.4
Duration of PROM	Not recorded	65	87.8
	<6 h	1	1.4
	≥6 h	8	10.8
Time of antibiotic administration	Before OR	2	2.7
	Before skin incision	72	97.3
Blood transfusion	No	65	87.8
	Yes	9	12.2
Postoperative hospital duration	<3 days	63	85.1
	≥3 days	11	14.9
Treatment received	No treatment	2	2.7
	Local treatment	5	6.8
	Systemic antibiotics	66	89.2
	Systemic antibiotics + surgery	1	1.4

PROM = premature rupture of membranes; OR = operation room.

**Table 4 healthcare-12-01474-t004:** Logistic regression for risk factors related to SSI in cesarean patients.

Dependent Variable = SI		OR (95% CI)	*p* Value
**Independent variables**	Age	0.67 (0.15–2.93)	0.591
	Nationality	0.13 (0.00–4.61)	0.265
	BMI ≥ 30	1.22 (0.28–5.39)	0.796
	Parity > 4	0.70 (0.12–4.20)	0.697
	Gestational age	1.71 (0.29–10.27)	0.556
	Labor before operation	0.72 (0.10–5.24)	0.743
	Status membrane	0.63 (0.11–3.74)	0.610
	Diabetes mellitus	10.76 (1.14–101.70)	0.038
	Hypertension	0.11 (0.01–2.69)	0.177
	Sickle cell disease	1.10 (0.08–15.17)	0.944
	Cardiovascular diseases	0.11 (0.00–4.34)	0.235
	No comorbidities	1.35 (0.19–9.91)	0.765
	Type of operation	2.02 (0.18–23.27)	0.572
	Duration of operation > 1 h	3.54 (1.49–7.17)	0.002
	Anesthesia	0.64 (0.17–2.31)	0.491
	Duration of PROM	0.57 (0.17–1.91)	0.364
	Blood transfusion	1.50 (0.19–11.68)	0.699

## Data Availability

The data presented in this study are available on request from the corresponding author.

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
