# Peer review of "Incidence of Surgical Site Infection Following Cesarean Section and Its Associated Factors in a Hospital of the Eastern Region, Saudi Arabia: A Retrospective Cohort Study"

_healthcare, 2024, doi:10.3390/healthcare12151474_

Round 1

Reviewer 1 Report

Comments and Suggestions for Authors

Dear authors,

The subject  is relevant and the three aims of the study are well described. 

First aim is to describe the incidence of SSI post CS. This aim is covered but mention that this is a monocentric study, outline a geographical location of the hospital, calculate the yearly number of births as a denominator.

Second aim is to identify risk factors. This aim is covered but mention the risks of small numbers statistics, show CI and Odds ratios and elaborate deeper on the risk factors:

- which preoperative antibiotics are the standard in your hospital, what about timely administration of these antibiotics, protocol for skin preparation protocol is available?, which antiseptics are used, where is the surgery performed (maternity ward or operation room?), is your hospital a teaching hospital,...

- Was  smoking  registered as a possible risk factor?

- CDC distinguishes between different types of SSI, from superficial to deep. Was this parameter recorded?

- Microbiology and the antibiotic resistance problem are briefly mentioned. Were MDRO isolated, was Staphylococcus aureus the most frequently encountered pathogen, were anaerobes involved?

The third aim is to scrutinize management protocols. This aim is not covered properly yet. Please elaborate on possible quality improvement projects (care bundles/ evidence based intervention protocols) that would work in your specific surroundings.

Comments on the Quality of English Language

line 49: Centers for Disease Control

line 72: play a fundamental role

line 166-167: The postoperative hospital stay was < 3days for most patients, while 14,9% had an extended stay of ....

Table 4: distinguishing between gram positive and gram negative is not sufficiently relevant. Omit this part of the table or differentiate properly (see my ealier remark on this subject)

Table 2 and 5: medically free   = none/ no comorbidities

Author Response

Summary

Point-by-point response to Comments and Suggestions for Authors

Comments 1: [First aim is to describe the incidence of SSI post CS. This aim is covered but mention that this is a monocentric study, outline a geographical location of the hospital, calculate the yearly number of births as a denominator..]

Response 1:

·       The geographical location of the hospital is already mentioned in materials and methods section, which is in the Eastern Region of Saudi Arabia. (Line 104).

·       The yearly number of births is used as a denominator Line (148), and also added in table 1.

Comments 2: [Second aim is to identify risk factors. This aim is covered but mention the risks of small numbers statistics, show CI and Odds ratios and elaborate deeper on the risk factors:.]

Response 2: Thanks for this suggestion. In fact as revealed by the statistical analysis, only diabetes and surgery time of more than one hour are the significant risk factors related to SSI. We thought it would be better to focus on the significant results rather than the small non-significant statistics although some brief discussion about this is mentioned in line 313 to 316.

Comments 3: [- which preoperative antibiotics are the standard in your hospital (1), what about timely administration of these antibiotics (2), protocol for skin preparation protocol is available? (3), which antiseptics are used (4), where is the surgery performed (maternity ward or operation room?) (5), is your hospital a teaching hospital (6),...

Response 3: All the above queries are done as per a valid hospital protocol.

1.     A first-generation cephalosporin “cefazolin” is the usually preoperative antibiotic used.

2.     The antibiotic is given within 60 minutes of skin incision

3.     Yes, a protocol for skin preparation is available

4.     The antiseptics used are as per protocol including chlorhexidine, and others. 

5.     The surgery is performed in operation room. 

6.     No, our hospital is not a university teaching hospital. It is a 330-bedded general hospital, and it is a certified centre for 12 medical and surgical residency training programs. It is already mentioned in materials and methods section. Line 104

Comments 4: [Was  smoking  registered as a possible risk factor?.]

Response 4: No, it was not. Smoking is a rare habit among women in the studied community.

Comments 5: [CDC distinguishes between different types of SSI, from superficial to deep. Was this parameter recorded?.]

Response 5: Yes, the SSI types was based on CDC guidelines, and it is already mentioned (Line 196).

Comments 6: [Microbiology and the antibiotic resistance problem are briefly mentioned. Were MDRO isolated, was Staphylococcus aureus the most frequently encountered pathogen, were anaerobes involved?]

Response 6: Thanks, done (Line 199-203)

Cultures conducted indicated various results: no growth was observed in 29.7% of cases, gram-positive bacteria were identified in 33.8% of cases, gram-negative bacteria were found in 25.7% of cases, and mixed growth in 10.8% of the cases. Of the total isolates, Staphylococci constituted 44.2%, Streptococci 23.1%, anaerobes 3.8%, and other gram-negative species 28.8%.

Comments 7: [The third aim is to scrutinize management protocols. This aim is not covered properly yet. Please elaborate on possible quality improvement projects (care bundles/ evidence based intervention protocols) that would work in your specific surroundings.]

Response 7: Thanks for this suggestion. In fact, some of this point is already elaborated in the discussion part (line 331-339).

Also we added these sentences (line 340-343): “Some quality improvement projects to decrease SSI rate include optimizing pregnant general health from the beginning of pregnancy with early antenatal care, managing chronic diseases that affect pregnancy outcome such as diabetes, and any emergency CS should be approved by a gynecology consultant whenever possible.

Response to Comments on the Quality of English Language

Point 1: line 49: Centers for Disease Control.

Response 1:  Thanks, done (line 64). 

Point 2: line 72: play a fundamental role.

Response 2:  Thanks, done (line 87).

Point 3: line 166-167: The postoperative hospital stay was < 3days for most patients, while 14,9% had an extended stay of ....

Response 3:  Thanks, done (line 190-191).

Point 4: Table 4: distinguishing between gram positive and gram negative is not sufficiently relevant. Omit this part of the table or differentiate properly (see my ealier remark on this subject).

Response 4:  Thanks, this part is removed, and the rest of this table is combined with table 3.

Point 5: Table 2 and 5: medically free   = none/ no comorbidities.

Response 5:  Thanks, done. 

Reviewer 2 Report

Comments and Suggestions for Authors

Please provide a STROBE checklist and Figure on study workflow.

Please list the sample size calculation, including the incidence used and the power.

Given the fact that there are a large number of patients without SSI, why was this study not conducted in a typical retrospective cohort manner, i.e., with negative and positive cases? Or match control study?

Table 1: Despite a steady number of c-sec cases, the incidence of SSI in year 2020 was quite low as compared with the others. Please further discuss

Table 2: Was there any socioeconomic data collected (income, occupation, education etc.)? How about the patients’ psychological status? Pain profiles before and after delivery are also not indicated.

Tables 3 and 4 can be combined.

Comments on the Quality of English Language

As above.

Author Response

Point-by-point response to Comments and Suggestions for Authors

Comments 1: [Please provide a STROBE checklist and Figure on study workflow]

Response 1: A STROBE checklist is already done and attached.

Comments 2: [Please list the sample size calculation, including the incidence used and the power]

Response 2: In fact, we used all the data set and included all cesarean section procedures performed during the 5-year studied period as mentioned in the methods section, from which we aimed to calculate the incidence rate of SSI. Therefore, there is no need to estimate the sample size as we did not use a sample of the whole data set.

Comments 3: [Given the fact that there are a large number of patients without SSI, why was this study not conducted in a typical retrospective cohort manner, i.e., with negative and positive cases? Or match control study?]

Response 3: Thanks for this suggestion; however, we aimed to just examine the SSI rate following cesarean section and risks associated. If we would like to include control cases, we should include deliveries without cesarean section. In such a case, the comparisons would not be appropriate.

Comments 4: [Table 1: Despite a steady number of c-sec cases, the incidence of SSI in year 2020 was quite low as compared with the others. Please further discuss..]

Response 4: The drop in the rate of SSI in year 2020 probably resulted from the fact that our hospital was a COVID-19 treatment center, and some pregnant women infected with COVID-19 were aggressively treated with broad-spectrum antibiotics in the critical care unit, and most of them had their pregnancies terminated by CS to save their lives. For this reason, the rate of CS in year 2020 was the highest (20.4%) compared to other years. (Line 302-308)

Comments 5: [Table 2: Was there any socioeconomic data collected (income, occupation, education etc.)? How about the patients’ psychological status? Pain profiles before and after delivery are also not indicated..]

Response 5:

·       Socioeconomic data unfortunately are not available in the hospital health information system (HIS).

·       For “the patients’ psychological status”, it was not part of our study scope; therefore, it was not included.  

·       For “pain profiles”, all the patients were given pain killers according to the hospital protocol, and it was not part of our study scope; therefore, it was not included.

Comments 6: [Tables 3 and 4 can be combined..]

Response 6: Thanks, done

Round 2

Reviewer 1 Report

Comments and Suggestions for Authors

Dear authors,

Thank you for the thorough revision. You answered my questions and the discussion is improving significantly.

Some final text editing:

L90: 300-bed hospital

L91: ...of Saudi Arabia, using a retrospective dataset on a cohort of women...

L254: ...beyond the 1-hour duration of surgery...

L255: ...between longer procedures and a higher...

Comments on the Quality of English Language

see minor text editing comments in above paragraph.

Author Response

Response to Comments on the Quality of English Language

Point 1: L90: 300-bed hospital.

Response 1:  Thanks, done (line 90). 

Point 2: L91: ...of Saudi Arabia, using a retrospective dataset on a cohort of women....

Response 2:  Thanks, done (line 91).

Point 3: L254: ...beyond the 1-hour duration of surgery...

Response 3:  Thanks, done (line 254).

Point 4: L255: ...between longer procedures and a higher...

Response 4:  Thanks, done (line 255).

Reviewer 2 Report

Comments and Suggestions for Authors

No further comments. Thanks.

Comments on the Quality of English Language

N.A.

Author Response

Thank you very much for taking the time to review the revised manuscript.